# Dichlorodiphenyltrichloroethane and the Adrenal Gland: From Toxicity to Endocrine Disruption

**DOI:** 10.3390/toxics9100243

**Published:** 2021-10-01

**Authors:** Ekaterina P. Timokhina, Valentin V. Yaglov, Svetlana V. Nazimova

**Affiliations:** A.P. Avtsyn Research Institute of Human Morphology, 3 Tsyurupy Street, 117418 Moscow, Russia; vyaglov@mail.ru (V.V.Y.); pimka60@list.ru (S.V.N.)

**Keywords:** endocrine disruptor, DDT, adrenal gland, mineralocorticoids, glucocorticoids, sex hormones, epinephrine, morphogenesis, transcriptional regulation

## Abstract

Endocrine disruptors are exogenous compounds that pollute the environment and have effects similar to hormones when inside the body. One of the most widespread endocrine disruptors in the wild is the pesticide dichlorodiphenyltrichloroethane (DDT). Toxic doses of DDT are known to cause cell atrophy and degeneration in the adrenal zona fasciculata and zona reticularis. Daily exposure in a developing organism to supposedly non-toxic doses of DDT have been found to impair the morphogenesis of both the cortex and the medulla of the adrenal glands, as well as disturbing the secretion of hormones in cortical and chromaffin cells. Comparison of high and very low levels of DDT exposure revealed drastic differences in the morphological and functional changes in the adrenal cortex. Moreover, the three adrenocortical zones have different levels of sensitivity to the disruptive actions of DDT. The zona glomerulosa and zona reticularis demonstrate sensitivity to both high and very low levels of DDT in prenatal and postnatal periods. In contrast, the zona fasciculata is less damaged by low (supposedly non-toxic) exposure to DDT and its metabolites but is affected by toxic levels of exposure; thus, DDT exerts both toxic and disruptive effects on the adrenal glands, and sensitivity to these two types of action varies in adrenocortical zones. Disruptive low-dose exposure leads to more severe affection of the adrenal function.

## 1. Endocrine Disruptors

Endocrine disruptors are exogenous compounds found in soil, water, air, and food. They produce hormone-like effects once they enter the body, even in very low doses, and disrupt the endogenous hormonal homeostatic mechanisms of regulation of the vital processes of living organisms. Endocrine disruptors are a global problem [1,2,3]. The term “endocrine disruptors” was introduced into the scientific literature in 1993 [4]. Shortly after the Endocrine Society published documents such as the 2012 Statement of Principles titled “Endocrine-Disrupting Chemicals and Public Health Protection”, letters were sent to the European Commission (March 2013) and the Secretariat for the Strategic Approach to International Chemicals Management (June 2013) calling for the introduction of an evidence-based approach to endocrine disruptors, which further contributed to raising awareness of these compounds and improving the understanding of the problem [5]. Endocrine disruptors include various classes of anthropogenic chemicals, such as pesticides (DDT and its metabolites), polychlorinated biphenyls [6,7,8,9], bisphenol A [10,11], polybromide diphenyl ethers [12,13,14], phthalates [15]; and other compounds, such as hormone-like substances of plant origin, which are contained in food [16,17]; various compounds used in the production of consumer and plastic goods; and other industrial environmental pollutants [18,19]. An elevated incidence of endocrine and immune disorders and cancers, particularly in childhood, as well as quicker onset of puberty and the impairment of reproductive functions point to endocrine and anthropogenic factors rather than just genetic factors [20,21,22,23].

## 2. Dichlorodiphenyltrichloroethane (DDT)

One of the most common endocrine disruptors found in the environment in both organisms and food is the pesticide dichlorodiphenyltrichloroethane (DDT). DDT is a contact insecticide affecting the insect’s nervous system. The toxicity level can be appreciated by the fact that fly larvae die after being exposed to a dose of less than 1 ng of DDT. During World War II, the use of DDT against malaria vectors saved millions of people from malaria, which was noted in the World Health Organization (WHO) report in 1973 [24]. Since the 1960s, about 400,000 tons of DDT have been used annually around the world, with 70–80% of that being used in agriculture. The relatively low acute toxicity for humans and animals and the low price (0.6 $/kg) of DDT have facilitated its intensive and unrestricted use [5]; however, its negative effects were soon recognized. For example, it has been proven that DDT has a toxic effect on the microbial flora of sea and river water, fish, amphibians, and birds. In 1970, Sweden was the first country to ban the use of DDT. The reasons for this were the persistence, bioaccumulation, and carcinogenicity of DDT [25,26,27]. In 2006, WHO decided to continue the use of DDT for malaria control in 12 countries around the world. Among them are India, North Korea, and some Southern African countries [28].

Today, 4000 to 5000 tons of DDT are used annually around the world to combat malaria and visceral leishmaniasis. India is currently the largest DDT consumer. In recent years, the production of this insecticide has even increased in some countries, such as India, North Korea, and China [29,30,31]. Due to its high resistance to decomposition, various doses of DDT and its metabolites are reported across all continents and in all oceans [32,33,34,35]. Its high solubility in fats and low solubility in water are the reasons for the DDT retention in adipose tissue. This fact was clearly demonstrated in a study of the distribution of DDT and its main metabolites DDD and DDE in young rats. It had already been reported that DDT and its metabolites accumulate in the liver, thymus, and brain, although the highest concentrations can be found in adipose tissue [36,37,38]. Most people around the world are still exposed to DDT through food. The highest levels of contamination are found in meat, fish, poultry, eggs, cheese, butter, and milk, since DDT easily accumulates in animal fat. It remains a widespread food contaminant that can be found in significant in areas where DDT production and use continue, as well as in areas where it was previously produced [39,40].

## 3. History of Investigations on Affection of the Adrenal Glands by Subtoxic and Toxic Doses of DDT

Investigations on the effects of DDT on the adrenal glands began in the 1940s. These studies included evaluations of the structural and functional changes to the adrenal cortex in various animal species. It should be noted that these studies only investigated toxic and sublethal doses. The ability of the DDT metabolite DDE to cause degeneration, cell death, and atrophy of the cortex was found in dogs [41,42], although further studies showed that these effects are typical only for dogs, while in rats, mice, rabbits, and monkeys, such manifestations were not observed [43]. The administration of DDT to Japanese quails caused increased mass of the adrenal gland, as well as expansion of the cortex. At the same time, the size of the nuclei of the cortical and medullary cells did not change. The lipid composition of the adrenal cortex also remained unchanged. The observed changes were thought to occur due to adrenocortical hyperplasia [44]. Studies of o,p’-DDT have shown that a dose of 50 mg/kg causes cell atrophy and degeneration in the zonae fasciculata and reticularis of the human adrenal cortex [45]; thus, it was found that the toxicity mechanism of o,p’-DDT is different from that of DDE, although their metabolisms are partly the same. Based on the data obtained, DDE isomers were introduced in the treatment of hypercortisolism and adrenal tumors in humans. The administered doses ranged from 7 to 285 mg/kg/day, although the therapeutic effect was observed at 100 mg/kg/day administered over many weeks [46,47]. This treatment was accompanied by severe physical and mental disorders, which decreased after the drug was no longer administered. There have been few studies on the effects of DDT and its residues on the zonae glomerulosa and reticularis, which were often contradictory. Specifically, some studies showed decreased aldosterone production when exposed to o,p-DDD and methylsulfonyl-DDE and increased androstenedione synthesis when exposed to methylsulfonyl-DDE but not o,p-DDT [48], while in other studies it was found that these compounds, when incubated with sections of the adrenal glands, do not bind either with cells of the zona glomerulosa or with cells of the medulla [49]. These results contradict the previously obtained data on the mechanisms of action of DDT as an insecticide. It is known that it has neuroparalytic action and affects both neurons and glial cells present in the medulla [50]. A few researchers have attempted to link the effects of DDT on the body with the development of Parkinson’s disease, based on the higher content of DDT metabolites in the brain tissues of the deceased who suffered from this disease [51]. In vitro experiments have shown that subtoxic doses of DDT metabolites increase the release of dopamine from synaptosomes and reduce its reuptake due to a decrease in the membrane dopamine transporter, as well as suppression of the vesicular monoamine transporter in a neurogenic lineage [52].

It is known that the adrenal cortex, in addition to mineralocorticoids and glucocorticoids, also secretes sex hormones. In this regard, the analysis of the effects of DDT and its metabolites on the endocrine function of the gonads that produce steroid hormones is of particular interest; however, in the literature, the information on the effects of DDT and its metabolites on steroidogenesis and reproductive function is ambiguous and even contradictory in some respects. The earliest studies indicated that DDT causes estrogen-like effects. These are manifested in the suppression of the growth of the testes and the development of secondary sexual characteristics in young male chickens. It has been shown that changes in the testes are visible not only in the seminiferous tubules but also in the interstitial tissue [53,54]. Studies carried out on two groups of male rats receiving DDT with food at doses of 50 and 100 mg/kg of body weight showed dose-dependent decreases in the testes and sperm motility [55]. At the same time, there was a decrease in the mass of seminal vesicles in addition to a decrease in testosterone production. Increases in the concentrations of luteinizing and follicle-stimulating hormones were noted in the serum. The authors regarded these changes to be a result of DDT’s action on the organs of the male reproductive system. At the same time, when DDT was administered in doses that had gonadotropic effects on the testes of male chickens and rats, many of them died on the first day of the experiment; thus, by the beginning of the 2010s, the effects of subtoxic and toxic doses of DDT on the body were well-studied, although there was almost no research on the effects of exposure to low doses of DDT.

## 4. Studies on the Effects of Low Doses of DDT on the Adrenal Glands

Since endocrine disruptors are agonists and antagonists of natural hormones, studies of the chemical interactions of DDT and the overall effects of DDT and its metabolites on the endocrine organs of animals and humans are of particular relevance. At the same time, it is known that during pregnancy, DDT and its metabolites can penetrate the placental barrier and affect the developing fetus [56,57,58,59]. The disruptor and its metabolites are also found in breast milk [60,61]. Accordingly, exposure to low doses of DDT begins at the embryonic stage; therefore, it is relevant to study constant low-dose exposure to DDT during all stages of the body’s development.

There is also ongoing research on the effects of endocrine disruptors on the transcriptional control of morphogenetic processes in endocrine organs [62,63].

### 4.1. Disorders of Adrenal Gland Secretion Induced by Low Doses of DDT

#### 4.1.1. Disruption of Adrenal Medulla Hormone Secretion by Low-Dose Exposure to DDT

Histological examination of the adrenal glands exposed to low doses of DDT revealed morphological and functional changes resulting from the redistribution of the venous outflow from the medulla to the cortex, due to the presence of anastomoses among their microcirculatory beds [64]. It is known that this redistribution occurs during the acute stress response, when it is necessary to intensify the transport of catecholamines to the liver through the portal vein system. This manifests in vasodilation of the microcirculatory vessels of the zona glomerulosa and outer zona fasciculata, stasis of erythrocytes, and even rupture of capillaries [65]. Evaluation of the epinephrine concentration in the systemic circulation and investigation of the secretory processes in the rat adrenal medulla revealed decreased functional activity of chromaffin cells, both in puberty and post-pubertal periods [64]. In rats that were developmentally exposed to low doses of DDT, pronounced and prolonged blood overflow and hemorrhages in the zona glomerulosa and outer zona fasciculata were associated with decreased catecholamine blood levels; thus, the redirection of blood to the portal vein through the cortex was an attempt to compensate for the insufficient production of catecholamines by the medulla, aimed at acceleration of gluconeogenesis in hepatocytes [64]. Further studies showed significant suppression of the thyrozine hydroxylase production in chromaffin cells after prolonged exposure to low doses of DDT [66]. These investigations clearly demonstrated the ability of low-dose exposure to DDT to change the cytophysiology of chromaffin cells and disruption of the adrenal medulla function.

#### 4.1.2. Disruption of Adrenal Cortex Hormone Secretion by Low-Dose Exposure to DDT

Low-dose exposure to DDT causes significant changes in the functioning of the adrenal cortex in rats. Developmental exposure resulted in a moderate increase in serum corticosterone levels and a significant decrease in the aldosterone concentration [67,68]. The concentration of female sex steroids produced in the zona reticularis, especially of estradiol, was significantly reduced [67,68]. Due to the sensitivity of zona reticularis and zona glomerulosa cells to ACTH, there was increased production of corticosterone via feedback mechanisms in the synthesis of hormones in ACTH-dependent zones [69,70,71,72]. An increase in the concentration of progesterone and its main metabolite, corticosterone, in addition to reduced production of mineralocorticoids and sex steroids, indicates a stimulating effect of the endocrine disruptor on steroidogenesis in the cells of the zona fasciculata. A distinctive feature of rats exposed to low doses of DDT in prenatal and postnatal development was an increase in the level of 17-hydroxyprogesterone, whose concentration in the blood serum of rats is normally extremely low [68]. This suggests that the effect of the disruptor does not reduce the hydroxylation of progesterone; therefore, a decrease in the production of sex steroids is associated with the later stages of their synthesis dysfunction [68], meaning the revealed changes are due to both the direct disruptor effect of DDT and the reaction of the hypothalamus–pituitary complex.

For this reason, the timing of exposure to the disruptor is an important factor. In rats exposed to disruptive doses of DDT starting from the day they were born, the concentrations of steroid hormones in the systemic circulation were significantly different from those of rats exposed prenatally. During puberty, postnatally exposed rats showed a decreased level of corticosterone and increased levels of aldosterone, estradiol, estrone, and testosterone [68]; thus, changes in the production of steroid hormones in postnatally exposed rats were diametrically opposite to those exposed both prenatally and postnatally. A common feature was that the concentrations of glucocorticoids and mineralocorticoids, as well as sex hormones, were directly correlated. This also suggests a regulating effect of ACTH. The decrease in the concentration of corticosterone was accompanied by the decrease in the level of its precursor progesterone, which indicates the suppression of steroidogenesis in the cells of the zona fasciculata. At the same time, the level of 17-hydroxyprogesterone did not differ from that of the control group, which in addition to an active conversion into sex steroids, also indicates that DDT does not interfere with the hydroxylation of progesterone. The level of glucocorticoid production was the only parameter that returned to normal levels after puberty. In rats exposed both prenatally and postnatally, the antiandrogenic effect of DDT was more pronounced, leading to an increase in estradiol production [73].

### 4.2. Changes in the Fine Structure of Adrenal Cells after Low-Dose Exposure to DDT

#### 4.2.1. Ultrastructural Changes of Chromaffin Cells

Adrenal chromaffin cells of pubertal rats exposed to DDT prenatally and postnatally demonstrated decreased numbers of mitochondria, especially under the plasmalemma. Another observation was a lower number of secretory granules containing epinephrine and reduced size of the mitochondria in norepinephrine-producing cells [74,75]. In the rats exposed to DDT from the first day after birth, destructive changes in mitochondria and a decrease in their number were observed, as well as a decrease in the number of secretory granules in the cytoplasm. The revealed changes were most pronounced in norepinephrine-producing cells. In the rats exposed to DDT both prenatally and postnatally, increased numbers of primary lysosomes in adrenal chromaffin cells and focal hemorrhages were found [76]. In the rats exposed to DDT postnatally only, the signs of increased synthetic activity and the formation of secretory granules, but not their release, were revealed [76]. This suggested that DDT disrupts the mechanisms of secretory material release from chromaffin cells due to a decrease in the number of mitochondria involved in the exocytosis of granules [74]. A comparison of blood catecholamine concentrations and the ultrastructure of chromaffin cells in rats exposed to DDT both pre- and postnatally vs. only in the postnatal periods showed that the disruption already begins during prenatal development. This leads to earlier onset of changes and the development of compensatory reactions, and accordingly a higher level of catecholamines after puberty. Consequently, prenatal exposure to DDT affects the rate of chromaffin tissue production in adrenals more than secretory machinery in the chromaffin cells.

#### 4.2.2. Ultrastructural Changes of Zona Glomerulosa Cells

Unlike chromaffin cells, adrenocortical cells do not store synthesized hormones in the cytoplasm within secretory granules. The main subcellular compartments for the assessment of steroid-producing cell activity are the mitochondrial apparatus and lipid inclusions in the cytoplasm. It is known that the structure of mitochondria is one of the main indicators of secretory activity of corticosterocytes [77]. The study of secretory processes in the adrenal zona glomerulosa in rats exposed to DDT pre- and postnatally showed the typical signs of hypofunction in the pubertal period—a decrease in cell size, a sharp increase in lipid inclusions in the cytoplasm, and a decrease in the percentage of mitochondria with a swollen matrix [75]. Consequently, a decrease in aldosterone production in rats was associated with a decrease in the steroidogenic activity of corticosterocytes. This decrease was largely due to microcirculatory disorders leading to hypoxia, which is a known factor in the reduction of the functional activity of corticosterocytes [69].

After sexual maturation, rats show an increase in the production of aldosterone, in addition to a decrease in the size of the zona glomerulosa [78]. This happens due to partial reorganization of steroidogenesis, the mitochondrial apparatus; specifically, the replacement of large mitochondria with a larger number of smaller mitochondria [79]. Prenatal and postnatal exposure to low doses of DDT slowed the reorganization of mitochondria, and despite active growth of zona glomerulosa in the postpubertal period, the level of aldosterone in exposed rats remained low [67]. Exposure to DDT in the postnatal period only allowed more rapid reorganization of mitochondria, which allowed increased production of the hormone [80].

#### 4.2.3. Ultrastructural Changes of Zona Fasciculata Cells

Rats exposed to DDT both pre- and postnatally demonstrate different changes in the fine structure during pubertal age. In the outer zona fasciculata, dystrophic changes and cell death were registered, especially in areas with microcirculation disorders. In the inner zona fasciculata the cells were larger, had larger nuclei, and a greater number of mitochondria, including those with a swollen matrix, as well as a more developed endoplasmic reticulum [81]. On the other hand, in postnatally exposed rats, the cells were smaller and the edema of the mitochondrial matrix was more pronounced, which seemed to be the main reason for their size increase. Examination of the fine structure and serum levels of corticosterone showed that suppression of hormone production was due to microcirculation disorders and cell death in the outer part of the zona fasciculata. Adequate production of corticosterone was supported by compensatory upregulation of steroidogenesis due to increases in mitochondria numbers and activation of steroidogenic enzymes in the cells of the inner zona fasciculata [81]; thus, the changes in rats that were only postnatally exposed to DDT reflect an earlier stage, while in rats exposed to the disruptor in both pre- and postnatal periods, a later stage of changes in steroidogenesis occurs due to structural cell changes.

#### 4.2.4. Ultrastructural Changes of Zona Reticularis Cells

Exposure to low doses of DDT in the prenatal and postnatal periods resulted in low steroidogenic activity of reticularis cells, despite their enlargement and a significant increase in the number of mitochondria [82]. After puberty, an acceleration of mitochondrial rearrangement was noted in compensation for the slowdown in the development of the zona reticularis; thus, low doses of DDT affect both the development of the zona reticularis and the structural support of secretory processes in its cells [83]. The data obtained showed that the disruptive effect of low doses of DDT on steroidogenesis in adrenocortical zones is not selective, in contrast to the action of toxic doses, which cause cell necrosis only in the zona fasciculata [49].

## 5. Dysmorphogenetic Disorders in the Adrenal Glands Induced by Low-Dose Exposure to DDT

The ability of disruptors to exert hormone-like effects due to structural similarity with molecules of endogenous hormones can significantly alter morphogenetic processes, especially if the disruptor mimics the action of steroid hormones that have receptors in the nucleus and directly regulate the transcription of various genes. DDT, therefore, can disturb the onset and the rates of the main morphogenetic processes, such as proliferation, apoptosis, and differentiation, in adrenal glands of the embryo. Studies have identified several signaling pathways, the activation of which plays an important role in the prenatal and postnatal development of the adrenal glands. These are the canonical β-catenin/Wnt signaling and Sonic Hedgehog signaling, the two key pathways that regulate cell proliferation, growth, and differentiation [84]. The latter is most active in the embryonic period; after birth it regulates the renewal and differentiation of pluripotent cells in the cortex in cooperation with the canonical Wnt [85,86,87].

Wnt (wingless-like MMTV integration site) is a family of morphogenic signals acting through a highly conserved, ancient evolutionarily pathway involved in embryonic development, regeneration processes, and maintenance of the cellular composition of organs. There are three known Wnt signaling pathways: two non-canonical (planar cell polarity and Wnt/Ca^2+^) and canonical β-catenin/Wnt signaling [88,89]. Furthermore, β-catenin acts as a coactivator and transmitter of extracellular signals for the activation of target genes. In addition to its role as a transcription factor, β-catenin is also involved in the formation of intercellular contacts, together with α-catenin and E-cadherin. A few studies have shown that these supramolecular complexes are targets for canonical Wnt, while switches of the adhesive and transcriptional functions of β-catenin have also been revealed [90,91,92,93]. The roles of canonical Wnt signaling in the postnatal development of structural and functional zones of the adrenal cortex are different, even the though age dynamics of the expression of the β-catenin transcription factor, for which penetration into the nucleus activates Wnt signaling, are the same in the zona glomerulosa, zona fasciculata, and zona reticularis [94]. Wnt signaling plays a key role in the development and differentiation of cells in the zona glomerulosa [95,96]. In contrast, growth and regeneration in the zona fasciculata require inhibition of Wnt signaling, since it suppresses the differentiation of its cells [97,98,99]. The role of canonical Wnt signaling in the development and functioning of the zona reticularis is very poorly understood. Single reports have shown that zona reticularis cells also express β-catenin in the postnatal period and that the percentage of cells with nuclear localization of β-catenin during adrenal growth is stable, in contrast to the content in the outer cytoplasmic membranes [94].

The impact of low doses of DDT below maximum permissible levels in food from the first day of prenatal development changes the morphogenetic processes in the adrenal glands. The mechanism of these changes is a disruption of transcriptional regulation, mainly in terms of proliferative processes. Morphogenetic processes in the medulla are less sensitive to the prenatal effects of the disruptor. At the same time, the adrenal cortex demonstrates sensitivity to both prenatal and postnatal effects, especially in the zona glomerulosa and zona reticularis. The zona fasciculata is less susceptible to the dysmorphogenetic action of low doses of DDT and its metabolites, in contrast to the action of toxic doses. Destructive and reparative processes in the rat adrenal cortex during puberty to a large degree are the result of microcirculation disorders. The disrupting affection from the beginning of the prenatal period causes the more rapid development of trophic cell disorders in the outer part of the zona fasciculata than after postnatal exposure, promoting a reactive increase in secretory activity in the deeper layers and then an increase in the number of mitochondria as a compensatory change to the disrupting effects of DDT. This is facilitated by the suppression of canonical Wnt signaling [80,100,101]. Prenatal exposure to the disruptor leads to significantly retarded development of the zona reticularis and zona glomerulosa. The relative hyperplasia of the zona glomerulosa, which develops after puberty, indicates a slowdown in its growth, which is due to the inhibition of the canonical Wnt signaling by DDT [100,102]. In the zona reticularis, the rate of development slows down to a greater extent, as evidenced by the lower degree of its development both during and after puberty [101]. In contrast to the zona glomerulosa and zona fasciculata, DDT increases the production of β-catenin and its content in the outer membranes of reticularis cells, but not translocation into the nucleus [103]. In addition to the canonical Wnt signaling, the dysmorphogenetic effect of DDT implicates disruption of age-related dynamics in the expression of Oct4 and Shh factors responsible for maintaining cell pluripotency and transdifferentiation, which also affect the levels of hormone production and reduce the regenerative potential of the cortex [103,104].

As such, the effect of low, disruptive doses of DDT on a developing organism causes changes in the postnatal morphogenesis of the adrenal cortex and medulla in rats and disrupts their secretory activity both during puberty and in adulthood.

## 6. Comparison of the Effects of Exposure to Toxic and Disruptive Doses of DDT

Low-dose exposure to DDT in prenatal and postnatal periods causes a lag in the development of the adrenal zona glomerulosa and zona reticularis and the acceleration of their development after reaching puberty, but does not affect the rate of development of the zona fasciculata [80]. These data show significant differences in the effects of toxic and disruptive doses on rodent adrenal glands (Figure 1 and Figure 2), since toxic doses of DDT induce degenerative and necrotic changes in the zona fasciculata, but not in the zona glomerulosa and zona reticularis [45,48,49,105,106]. Consequently, steroid-producing cells of the zona fasciculata are more sensitive to the toxic effects of DDT, while the zona glomerulosa and zona reticularis are more sensitive to the disrupting effects.

## 7. Conclusions

An important breakthrough in methodological approaches to the study of endocrine disruptors was a recognition of the failure of toxicological approaches; thus, the determination of threshold doses needs to be abandoned in favor of separating the toxic effects from the disruptive action of low doses. Hormones can act in concentrations ranging from ng/mL to pg/mL. Accordingly, endocrine disruptors cannot have a safe dose, and extremely low levels of exposure, corresponding to the background effects on the body, need to be studied. The significant differences in the effects of exposure to toxic and low doses of DDT on adrenal glands are obvious. Moreover, daily low-dose exposure over time results in more severe affection of the adrenal glands than prolonged exposure to subtoxic and toxic doses. Consumption of the endocrine disruptor DDT in doses below the maximum permissible levels in food products still changes the morphogenetic processes in adrenal glands. The mechanisms of these changes include impaired transcriptional regulation of primarily proliferative processes. The adrenal cortex demonstrates sensitivity to both the prenatal and postnatal effects of the disruptor, especially its zona glomerulosa and zona reticularis. The data obtained indicate the severity of disruption of adrenal growth and function due to low doses of DDT and its harmful effects both pre- and postnatally. Dysfunction of the adrenal glands and subsequent dysregulation of the physiological functions of organs and systems by their hormones may result in dysmorphogenetic and functional disorders. These disorders may trigger various pathological processes, primarily due to dysfunction of the immune, reproductive, and cardiovascular systems.

## Figures and Tables

**Figure 1 toxics-09-00243-f001:**
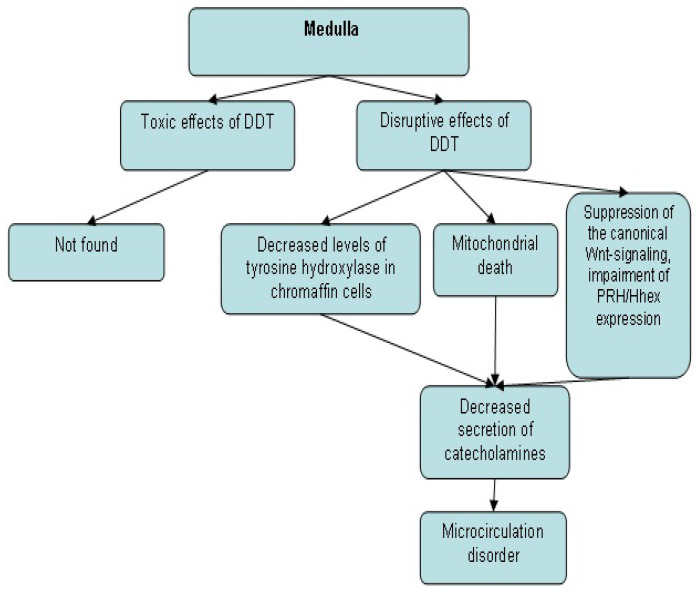
Changes in the morphogenesis and secretory activity of the adrenal medulla after exposure to toxic and disruptive doses of DDT.

**Figure 2 toxics-09-00243-f002:**
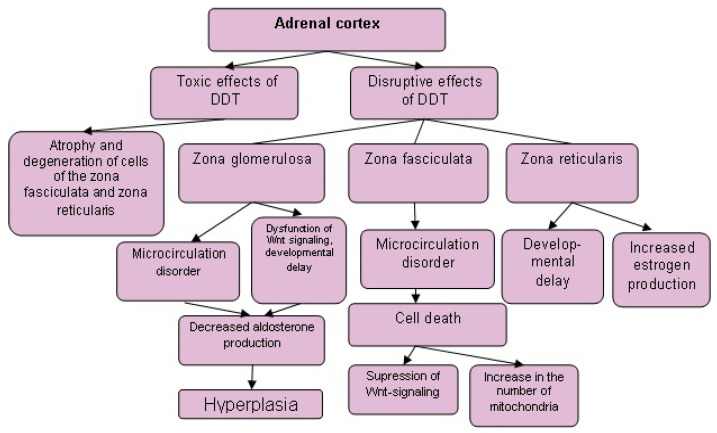
Changes in the morphogenesis and secretory activity of the adrenal cortex after exposure to toxic and disruptive doses of DDT.

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
