# Peer review of "Dichlorodiphenyltrichloroethane and the Adrenal Gland: From Toxicity to Endocrine Disruption"

_toxics, 2021, doi:10.3390/toxics9100243_

Round 1
Reviewer 1 Report
Review of manuscript entitled Dichlorodiphenyltrichloroethane and the adrenal gland: from toxicity to endocrine disuption (ms ID: toxics-1363383), by Thimokhina et al.
This review article attempts to summarize the described effects observed in the adrenal gland of various species after exposure to various quantities of DDT, including the discussion of such effects exerted at different ages.
The topic is not only interesting, but also it is extremely important, and the content of the manuscript should be eventually published. However, before the publication of the manuscript, there are some flaws to overcome by the authors.
There are two major problems with the paper:
- The English is extremely poor. There are a number of sentences or even paragraphs whose understanding is a struggle to the reader. This problem includes the incorrect use of words, as well as the incorrent sequence of words in many sentences. There is a sharp contrast between the scientific weight of the topic and the language in which it is presented.
- Besides the poor English, there is something the authors should correct, or better said, they should re-define: this is the classification of the effects of DDT. The authors sort the effects of DDT into either being toxic or being disruptive in the endocrine system. This classification is reflected from the very beginning of the paper (the title) till the end. However, endocrine disruption is also a way of toxicity. At the beginning of their description the authors present DDT effects as either being toxic (with detectable morphological malformations) or being disruptive (to the reader this comes as kind of effects on functions, or functional effects), however, there is no clear border between the two groups.
At the same time, morphological and functional conditions come together, therefore such a distinction between DDT effects is hard to accept and the authors get trapped in this logical ditch themselves in their Conclusion (lines 368-372). The reviewer strogly suggests a different kind of classification between DDT effects.
Other comments:
- The use of the term „dose” may not be always correct. When experimental results are cited in which biological responses to well known amounts of applied substance were described, the use of the term „dose” is certainly correct. When the discussed observations are cited from conditions where the exact amount of DDT hitting the organism was unknown, but its known or estimated amount in the macro or microenviroment was described, the term „concentration” is more correct. Briefly, amount-associated effects can be either dose- or concentration-dependent, but the two are not to be confused.
- between lines 311-338, the discussion of effects would be better if the authors would take the layers of the adrenal gland one by one, one at a time, instead of jumping back and forth between these layers. That way it would be much easier to follow for the reader.
Author Response
Dear Reviewer!
Thanks for your review.
1. English text revised for a better understanding.
2.We do not agree with determination of endocrine disrupting effects as toxic ones. Endocrine disruption affects regulation of cell development and function and does not cause cell death directly
Reviewer 2 Report
The review “Dichlorodiphenyltrichloroethane and the adrenal gland: from toxicity to endocrine disruption” by Timokhina et al. is well written and the topic is well defined. In this review, the authors provide an adequately literature about the the disruptive effects of low DDT doses on adrenal glands. Dichlorodiphenyltrichloroethane is a well-known insecticide and although the literature contains a plenty of papers dealing with the toxics effects on vertebrates and invertebrates as well as wildlife and human health, little is known on the effects of low DDT doses on organisms health. This review takes into account the effects of low, disruptive, DDT doses on adrenal glands and several of the significant papers on mainly this topic are published by the authors of the present review. The literature is update and the results of the published studies included in the current review are significant and discussed appropriately. The review is of the interest of the readership of journal and provides an advance towards the current knowledge. The English language is appropriate.
I provide some comments/suggestions, listed below, that may help the Authors to further improve the MS.
In this review the authors do not report the concentrations of DDT underlined in the various works. Please, I would suggest indicating the concentrations of toxic or low doses of DDT so that the readers of the journal can get an immediate information at least on the doses exerting a disruptive effect on the adrenal glands.
Keywords are informative and reflect the content of review. I would suggest ordering Keywords in alphabetical list.
Abstract helps readers to focus the aims and the main outcomes of the review. The paragraphs clearly exposes the research topic and related discrepancy and gaps of scientific literature. Conclusions are supported by the published papers.
L 84 please check the reference of Bloch et al. Cancer. 1956. It seems to me to be out of place.
L98-102 for an easy reading, please, split it in two sentences.
L 130-132 I would suggest to rephrase it for a better understanding of sentence
Both figures are well depicted and informative; however I would suggest to enlarge them for an easy reading.
Author Response
Dear Reviewer!
Thanks for your review.
The erroneous reference has been replaced. English text revised for a better understanding.
Round 2
Reviewer 1 Report
The revized version of the manuscript is very impressive, the English as well. As the reviewer stated during the first review, the content of the paper is very precious and should be eventually published. The is one major problem in the opinion of the reviewer: as also stated already, the concentration dependent effects of the endocrine disruptors cannot be classified and distinguished as "toxic" effects and "low-dose" effects. The authors argue against this opinion, but at the same time they state that long-time exposure to low-dose DDT may cause even tougher effects than the so-called toxic effects. All in all, when there is an effect that pushes life parameters out of homeostatic limits they cannot be considered as non-toxic. In other words, all such effect are toxic in one way or another, and that is why this reviewer suggested that the authors would re-name the two major concentration-dependent groups of effects. Again, the manuscript is of high quality, with the notion that the aforementioned classification of effects is unintelligible. If the authors still cannot accept the reviewer's request, I suggest the delegation of this decision to another referee, because I cannot be convinced about the correctness of the effect classification in question.
Author Response
Dear Reviewer!
Thanks for your review.
We still believe that the toxical and disruptor effects are fundamentally different.